# Simple Route to Synthesize Fully Conjugated Ladder Isomer Copolymers with Carbazole Units

**DOI:** 10.3390/polym11101619

**Published:** 2019-10-07

**Authors:** Shuang Chen, Feng Liu, Chao Wang, Jinghui Shen, Yonggang Wu

**Affiliations:** 1College of Chemistry and Environmental Science, Hebei University, Baoding 071002, China; chengshuangfff@163.com (S.C.); WC19940119@163.com (C.W.); sjhui0218@163.com (J.S.); 2College of Physics Science and Technology, Hebei University, Baoding 071002, China

**Keywords:** carbazole, ladder, BO embedded, isomer

## Abstract

Two isomer polymers, P3 and P6, with fully conjugated ladder structures are presented by simple synthetic routes. The well-defined structures of fully conjugated ladder polymers P3 and P6 were ensured by the high yields of every reaction step. The fully rigid ladder structures were confirmed by nuclear magnetic resonance (NMR), fourier transform infrared spectroscopy (FTIR), and photophysical test. Polymers P3 and P6 with bulky alkyl side chains exhibit good solution processability and desirable thermostable properties. After the intramolecular cyclization reaction, the band gaps of polymers P3 and P6 become lower (2.86 eV and 2.66 eV, respectively) compared with polymers P1 and P4. This initial study provides insight for the rational design of fully ladder-conjugated isomeric polymers with well-defined structures.

## 1. Introduction

Fully conjugated ladder polymers (cLPs) have drawn extensive attention because of their rigid structure with π-conjugated systems [1,2,3,4,5,6]. The unique structure of cLPs confers favorable physical, optical, and chemical properties [7,8,9,10]. Thus, cLPs are promising candidates as organic materials [11,12,13,14,15,16], such as organic solar cells, organic thin film transistors, and organic light emitting diodes. However, there still exist some challenges to synthesize new cLPs with the desired solubility and fewer structure defects, and researchers have tried various methods to solve these difficulties [6,17,18,19,20,21], such as introducing bulky alkyl side chains to improve solubility, and introducing new reactive groups using new synthetic methods to synthesize ideal conjugated defect-free polymers.

Recently, researchers have introduced different heteroatoms [8,22,23,24,25,26], such as N, B, S, P, and Si, to the fused π-conjugated structure. This can effectively regulate the structure with desirable physical and chemical properties. The B-N unit, which has strong electron-withdrawing properties, can be used to construct new materials with good charge transport and photoconductivity in optoelectronic applications [22,27,28,29,30,31,32,33,34]. Hence, the integration of the B-N fragment into the fused π-conjugated ladder system may be a useful strategy to construct new cLPs. Wu’s group [21,35,36,37] has conducted extensive work of this type. They once used the above strategy to synthesize a fused π-conjugated ladder compound and polymer [35]; the polymer exhibited good solution and thermal properties, with a UV-vis absorption peak at 455 nm; after the BN-embedded cyclization, the fused ladder-conjugated polymer presented a lower band gap.

In this work, we designed simple synthetic routes to synthesize two isomer polymers, P3 and P6, with BO-embedded fused π-conjugated ladder structure. The high yields of each synthesis step ensure the well-defined structure of the polymer with fewer structure defects. The BO-embedded fused π-conjugated ladder structures of P3 and P6 were confirmed by NMR, FTIR, and photophysical test. Their thermal, photophysical, and electrochemical properties were also studied in detail. The main purpose of this paper is to provide an insight for synthesizing defect-free cLPs.

## 2. Materials and Methods

### 2.1. Materials

Commercial chemicals were used without further purification. Tetrahydrofuran (THF) and dichloromethane (DCM) were distilled by a standard process before use. All the reactions were monitored by thin layer chromatography (TLC) with silica gel 60 F254 (Merck, Germany, 0.2 mm).

### 2.2. Instrumentation

^1^H and ^13^C NMR data were obtained on a Bruker AV600 spectrometer. The electrochemical behavior was recorded by cyclic voltammetry (Holland, Ivium Plus II) with a standard three-electrode electrochemical cell, including a glassy carbon working electrode, a Pt wire counter electrode, and an Ag/Ag^+^ (0.01 M in CH_3_CN) reference electrode. The CH_2_Cl_2_ solution contains 0.1 M tetrabutylammonium hexafluorophosphate (TBAPF6).The scan range was −1.4–1.7 V, and the scan rate was 50 mV·s^−1^. UV-visible absorption spectra were obtained on a Shimadzu UV-visible spectrometer model UV-2550 (Japan). Fluorescence spectra were obtained by a Shimadzu RF-5301PC fluorescence spectrophotometer (Japan). MALDI-TOF analyses were acquired with Bruker Daltonics Inc Autoflex III (US). Thermal gravimetric analysis (TGA) and differential scanning calorimetry (DSC) measurements were carried out on Perkin-Elmer Pyris 6 TGA (US) (10 °C/min, N_2_) and Perkin-Elmer Diamond DSC instruments (US) (10 °C/min, N_2_), respectively, to record TGA and DSC curves.

### 2.3. General Synthesis of Conjugated Ladder Polymers P3 and P6

The conjugated ladder-type copolymers were synthesized in three steps (Scheme 1). First, polymers P1 and P4 were synthesized by using Suzuki coupling conditions and Pd(PPh_3_)_4_ as a catalyst. Then, the ester group was converted into a hydroxy group. Last, phenylboron dichloride was added for the intramolecular cyclization reaction to give polymers P3 and P6. The detail synthesis for the reaction monomers, compounds, and polymers is provided as Appendix A.

## 3. Results

### 3.1. Synthetic Route Discussion

To ensure the synthetic route could successfully synthesize the defect-free fully π-conjugated ladder polymers, we first used model compounds to explore the synthetic routes. The compound S1 was synthesized in high yields above 96%. As seen in Scheme 2, the carbazole and phenol units of compound a2 were linked by a single bond; thus, the phenol-carbazole bond unit can have free rotation. The intramolecular cyclization may form three possible isomers including compounds S1, S2, and S3. However, the structure of the product was identified by NMR spectrum, MALDI-TOF, and FTIR (see Appendix A), and only one product, compound S1, was observed. The geometry optimizations of compound S1, S2, and S3 were managed within the framework of density functional theory (DFT) using B3LYP functional at the level of 6-31G(d) (Figure 1). Methyl units were used to replace branched side chains. The electronic and thermal free energies (E) of compound S1, S2, and S3 were −1679.65 KJ/mol, −1679.64 KJ/mol, and −1679.63 KJ/mol, respectively. The compound S1 had the lowest electronic and thermal free energies, indicating that S1 is more stable than S2 and S3 [38,39,40,41]. The high yield of the intramolecular cyclization reaction and deterministic structure indicates a high feasibility to synthesize the fully conjugated ladder polymers P3 and P6. The photophysical and electrochemical properties of the compounds were also studied (see Appendix A).

### 3.2. NMR Spectra of Polymers

The structures of the polymers were first identified by NMR (Figure 2). First, both the NMR spectra of P1 and P4 showed a clear peak around 2.44 ppm. This peak was assigned to the -COOCH_3_ protons. Then, when the ester group was converted into the hydroxy group, the peak around 2.44 ppm disappeared completely, and a new peak appeared around 5.30 ppm or 5.67 ppm. This new peak was assigned to the -OH proton. Last, after the intramolecular cyclization reaction, the peak around 5.30 ppm or 5.67 ppm completely disappeared, and a new peak appeared around 9.10 ppm. This new peak may be assigned to the protons of the benzene ring. Comparing the NMR spectra of P1, P2, P3, P4, P5, and P6, the completely disappeared peaks and newly formed peaks indicate that each step of the synthetic reaction has a high yield, and the structures of P3 and P6 have a well-defined structure with fewer defects.

### 3.3. Size-Exclusion Chromatography (SEC) Trace for Polymers

The molecular weight of polymers was investigated by size-exclusion chromatography (SEC) in THF (polystyrene as standard) (Figure 3), and the data are listed in Table 1. The *M_n_* values of P1, P3, P4, and P6 were 7.1 × 10^3^ g/mol, 7.8 × 10^3^ g/mol, 9.8 × 10^3^ g/mol, and 10.8 × 10^3^ g/mol, respectively, and the polydispersity index (*Ð*) was 1.49, 1.46, 1.59, and 1.40, respectively. The *M_n_* values of P3 and P6 can be recognized as being overestimated relative to the actual molecular weight due to the rigid structure. The planar structure of the polymers can increase the hydrodynamic radius in solution.

Usually, the fully π-conjugated ladder polymers showed limited solubility due to the rigid structure and strong intramolecular interactions. To solve this problem, bulky alkyl chains were introduced into the polymer structures. Due to the bulky alkyl chains, polymers P3 and P6 showed good solubility in common organic solvents such as chloroform, THF, and *N,N*-dimethylformamide (DMF).

### 3.4. Thermal Properties of Polymers

The thermal properties of copolymers were evaluated by thermal gravimetrical analysis (TGA) (Figure 4) and differential scanning calorimetry (DSC) (Figure 5). The decomposition temperatures (with a 5% mass loss) for polymers P1, P3, P4, and P6 were 381 °C, 363 °C, 351 °C, and 368 °C, respectively. In the heating and cooling DSC scans, there were no apparent glass transition processes or other thermal processes. This phenomenon suggested that the polymers have an amorphous structure. Also, their thermal properties were not influenced by the isomeric effect.

### 3.5. FTIR Spectra of Polymers

Furthermore, the intramolecular cyclization reaction was also confirmed by FTIR spectrum (Figure 6). First, the polymers P1 and P4 showed a -C=O stretch peak (around 1736 cm^−1^). After the dehydration reaction, the ester group changed into a hydroxyl group, with the loss of the -C=O stretch peak and a new -OH stretch peak (3542 cm^−1^ for polymer P2, 3455 cm^−1^ for polymer P4). The -C=O stretch peak completely disappeared, which indicates the high yield of the dehydration reaction. Last, comparing polymer P2 with P3, and P4 with P6, the -OH stretch peak completely disappeared, indicating the high yield of the intramolecular cyclization reaction and the less structure-defect for polymers P3 and P6.

### 3.6. Photophysical Properties of Polymers

The UV-vis absorption and fluorescence spectra of polymers were measured in a diluted THF solution (Figure 7a,b ). Polymers P1, P2, and P3 showed a similar UV-vis absorption peak around 330 nm (326 nm, 344 nm, and 326 nm, respectively). Clearly, the polymer P3 showed another UV-vis absorption peak around 390 nm, which is attributed to the extended π-conjugated structure after the B-O intramolecular cyclization reaction. The polymers P1, P2, and P3 exhibited maximum emission peaks at 398 nm, 409 nm, and 435 nm, respectively. The quantum yields (*Φ*_f_) of P1 and P3 were calculated as 0.66 and 0.96, respectively, with 9,10-diphenylanthracene (*Φ*_f_ = 0.95) as the reference. The higher *Φ*_f_ value of P3 can be ascribed to the more rigid structure and electron-donor properties when phenyl groups are introduced. The UV-vis absorption and fluorescence spectra of polymers in film were also measured. The UV-vis absorption peaks of polymers P1, P2, and P3 were around 337 nm, 342 nm, and 345 nm, respectively. Compared with the UV-vis absorption of polymers in THF solution, the UV-vis absorption of polymers in film did not show any shifts. Meanwhile, the emission peaks of polymers in film showed a small red shift compared with the emission peaks of polymers in THF solution.

Polymers P4 and P5 showed a single UV-vis absorption peak around 363 nm and 372 nm in THF solution, respectively. Polymer P6 showed two UV-vis absorption peaks around 380 nm and 435 nm, respectively. Compared with P4, the UV-vis absorption peak of P6 showed an obvious red shift. The polymers P4, P5, and P6 exhibited maximum emission peaks at 424 nm, 450 nm, and 476 nm, respectively. The quantum yields (*Φ*_f_) of P4 and P6 were calculated as 0.72 and 0.98, respectively, with 9,10-diphenylanthracene (*Φ*_f_ = 0.95) as the reference. Meanwhile, compared with the UV-vis absorption and emission of polymers in THF solution, the UV-vis absorption and emission of polymers in film showed a small red shift.

Compared with P3, the UV-vis absorption peaks of P6 in THF solution showed an obvious red shift, as the structure of P6 may show more rigid planar structures. The band gap of polymers P1, P3, P4, and P6 were 3.14 eV, 2.86 eV, 3.00 eV, and 2.66 eV, respectively. Compared with P1 and P4, P3 and P6 showed a lower band gap.

### 3.7. Electrochemical Properties of Polymers

The electrochemical properties of polymers were assessed using cyclic voltammetry and the Fc/Fc^+^ redox system as the internal standard (−4.80 eV). The cyclic voltammogram (CV) measurements are summarized in Figure 8 and Appendix A. All the data of polymers are list in Table 1 and Appendix A The HOMO values of polymers P1, P3, P4, and P6 were −5,52 eV, −5.62 eV, −5.56 eV, and −5.68 eV, respectively. The LUMO values of polymers P1, P3, P4, and P6 were −2.38 eV, −2.76 eV, −2.60 eV, and −3.02 eV, respectively.

## 4. Conclusions

Two isomer polymers P3 and P6 with fully conjugated ladder structures have been presented. Simple synthetic routes with a high reaction yield ensure the well-defined structures of fully conjugated ladder polymers P3 and P6 with fewer structure defects. The rigid structures were confirmed by NMR, FTIR, and photophysical test. Polymers P3 and P6 exhibited good solution processability. The *M_n_* values were 7.8 × 10^3^ g/mol and 10.8 × 10^3^ g/mol for P3 and P6, respectively. The polymers P3 and P6 showed two distinct UV-vis absorption bands. The HOMO values were −5.62 eV and −5.68 eV for P3 and P6, respectively. Compared with P1 and P4, P3 and P6 showed a lower band gap (2.86 eV and 2.66 eV, respectively). The TGA result showed that all the polymers had desirable thermostable properties. This initial study provides insight for the rational design of fully ladder-conjugated isomeric polymers with well-defined structures.

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
