# Peer review of "Simple Route to Synthesize Fully Conjugated Ladder Isomer Copolymers with Carbazole Units"

_polymers, 2019, doi:10.3390/polym11101619_

Round 1

Reviewer 1 Report

The manuscript report the synthesis and characterization of new fused conjugated ladder polymer. The prepared polymers have been investigated by different experimental methods and the data show that these polymers could have suitable applications in organic electronics field. Globally, the work has been correctly conducted. However, the quality of English has to be improved. There are many typomistakes in the document and herein I'll cite some examples. The manuscript lacks also the theoretical investigation. DFT can give more insight on the optoelectrochemical properties of the polymers and explain why only S1 was formed. 

other comments:

alky side chain => alkyl side chain (for instance: line 14, 29 etc)

intermolecular cyclization => intramolecular cyclization (many times, for instance: line 80, 83, 132, etc)

The sentence line 34 – 35: should be rewritten

Line 35, 37:  fuse ladder => fused ladder

Line 80 – 83: provide an explanation why only S1 was formed. DFT calculation can give more insight about this observation

Line 101: NMR solvent?

Line 112: common organic solution => common organic solvent

The part 3.6 photophysical properties:

Please provide the absorption and emission spectra in solid state and compare with the data obtained in solution. Figure 6. What is the nature of the broad band of P2 centered at 500 nm? Why we do not observe this band within P5?

Part 3.7 electrochemical properties

Figure 7 and line 168: the CV curves of the polymers are mentioned, but where are they? In the ESI fig S4: please indicate clearly how you determine the oxidation, reduction potentials for HOMO LUMO energy calculation (especially the reduction part to avoid confusing with the signal from the solvent)

Author Response

Reviewers’ comments (in black) and our response (in blue)

Reviewer: 1

Comments:

The manuscript report the synthesis and characterization of new fused conjugated ladder polymer. The prepared polymers have been investigated by different experimental methods and the data show that these polymers could have suitable applications in organic electronics field. Globally, the work has been correctly conducted. However, the quality of English has to be improved. There are many typo mistakes in the document and herein I'll cite some examples. The manuscript lacks also the theoretical investigation. DFT can give more insight on the optoelectrochemical properties of the polymers and explain why only S1 was formed.

Reply: We thank the reviewer for the positive recommendation and address the comments below.

There are many typo mistakes in the document and herein I'll cite some examples.

Reply: We are now modifying the spelling and grammar errors as the reviewer cite in the article, such as “alkyl side chain, intermolecular cyclization, fused ladder, and common organic solvent”.

The sentence line 34-35 should be rewritten.

Reply: The sentence line 34-35 has been rewritten. “B-N unit, which own strong electron withdrawing properties, can be used to constructed new materials. Those new materials can exhibit good charge transport and photoconductivity in optoelectronic application.”

Line 80 – 83: provide an explanation why only S1 was formed. DFT calculation can give more insight about this observation

Reply: The geometry optimizations of compound S1, S2 and S3 were managed within the framework of density functional theory (DFT) using B3LYP functional at the level of 6-31G(d).  Methyl units were used to replace branched side chains. The electronic and thermal free energies (E) of compound S1, S2 and S3 were -1679.65 KJ/mol, -1679.64 KJ/mol and -1679.63 KJ/mol, respectively. The compound S1 own the lowest electronic and thermal free energies, indicating S1 is more stable than S2 and S3[1-4]{Türker, 2018 #942}.

Figure 1. The optimal structure and total energy of compound S1, S2 and S3 based on B3LYP/6-31G(d) level by Gaussian 16.

Reference:

Kakkar, R.; Pathak, M.; Radhika, N.P. A DFT study of the structures of pyruvic acid isomers and their decarboxylation. Organic & Biomolecular Chemistry 2006, 4, 886-895, 10.1039/B516355B. Ghiasi, R.; Ara, T.J.; Hakimyoun, A.H. Spectroscopic studies and molecular orbital analysis on platinanaphthalenes and ring-fused B-N platinanaphthalenes. Russian Journal of Physical Chemistry A 2014, 88, 616-624, 10.1134/s0036024414040244. Türker, L. A DFT study on TNGU isomers and aluminized cis-TNGU composites. Defence Technology 2018, 14, 109-118, https://doi.org/10.1016/j.dt.2017.08.001. Liu, D.; Yu, B.; Su, X.; Wang, X.; Zhang, Y.-M.; Li, M.; Zhang, S.X.-A. Photo-/Baso-Chromisms and the Application of a Dual-Addressable Molecular Switch. Chemistry – An Asian Journal 2019, 14, 2838-2845, 10.1002/asia.201900600.

Line 101: NMR solvent?

Reply: The NMR spectrum of polymers recorded in CDCl3.

The part 3.6 photophysical properties: Please provide the absorption and emission spectra in solid state and compare with the data obtained in solution. Figure 6. What is the nature of the broad band of P2 centered at 500 nm? Why we do not observe this band within P5?

Reply: We provide the absorption and emission spectra in solid state and compare with the data obtained in solution.

We also retest the absorption of P2 and the absorption band centered at 500 nm was disappeared. This mistake may be caused by test instrument.

Part 3.7 electrochemical properties? Figure 7 and line 168: the CV curves of the polymers are mentioned, but where are they? In the ESI fig S4: please indicate clearly how you determine the oxidation, reduction potentials for HOMO LUMO energy calculation (especially the reduction part to avoid confusing with the signal from the solvent?

Reply: The cyclic voltammogram (CV) measurements were summarized in Figure 7 and Figure S4. All the result data of polymers were listed in Table 1 and Table S1. The HOMO energy and LUMO energy were calculated by EHOMO = -(Eox+4.8) eV, ELUMO = EHOMO+Egap. The equations were listed below the Table 1.

Reviewer 2 Report

The paper presents the synthetic route of Fully Conjugated Ladder Isomer Copolymers although the presentation of the synthesis and characterization methods described are insufficiently analysed or not clear so the the reader can easily follow them.

In some of the cases such as the description of the molecular weight determination by SEC both the terms SEC and GPC appear (Table 1). The authors should also take into consideration that the term polydispersity index (PDI) is not replaced by IUPAC by the term dispersity with the symbol Đ.

The language of the whole text needs improvement.

Author Response

Reviewers’ comments (in black) and our response (in blue)

Reviewer: 2

Comments: The paper presents the synthetic route of Fully Conjugated Ladder Isomer Copolymers although the presentation of the synthesis and characterization methods described are insufficiently analysed or not clear so the reader can easily follow them.

In some of the cases such as the description of the molecular weight determination by SEC both the terms SEC and GPC appear (Table 1). The authors should also take into consideration that the term polydispersity index (PDI) is not replaced by IUPAC by the term dispersity with the symbol Đ. The language of the whole text needs improvement

Reply: We thank the reviewer for the positive recommendation and address the comments below.

We descript the molecular weight determination by SEC. The polydispersity index was used the symbol Đ

Reviewer 3 Report

I believe that the novelty of the work is the synthesis of P3 and P6 isomers based on the preferential synthesis of a model compound S2. But effort to compare S2 with P3 and P6 has not been made. Further, the authors claim high yield of the reaction but did not report yield of the main reaction in the manuscript.

In Page 4, authors claim that P3 and P6 are soluble in organic solvents but their solubility data was never reported. They claim that solubility of these polymers in organic solvent is due to the presence of alkyl chains. Does that mean S2 is also soluble in organic solvents?

Overall, the manuscript is poorly written without sufficient discussion on the claims made. In addition to it, a lot of grammatical mistakes makes the readability even more difficult.

I do not recommend the publication of this manuscript in the current form. 

Author Response

Reviewers’ comments (in black) and our response (in blue)

Reviewer: 3

We thank the reviewer for the positive recommendation and address the comments below.

I believe that the novelty of the work is the synthesis of P3 and P6 isomers based on the preferential synthesis of a model compound S2. But effort to compare S2 with P3 and P6 has not been made. Further, the authors claim high yield of the reaction but did not report yield of the main reaction in the manuscript.

Reply: The model compound was compound S1 and was not compound S2. In this work, we use the model compound S1 to ensure the synthetic route can successfully synthesize the polymer P3 and P6. The properties of P3 and P6 isomers were studied in detail. Herein, we didn’t compare S1 with P3 and P6.

The high yield of the reaction was above 96%. We list the yield of the reaction at “3.1 Synthetic Route Discussion” and the yields of every reaction step were showed at “Supporting Information, Experimental Section”.

In Page 4, authors claim that P3 and P6 are soluble in organic solvents but their solubility data was never reported. They claim that solubility of these polymers in organic solvent is due to the presence of alkyl chains. Does that mean S2 is also soluble in organic solvents?

Reply: Due to the rigid backbones, fully conjugated ladder polymers often have strong π-π interactions that can limit their solubility. A typical method to solve this problem is to introduce bulky side chains to the backbone to disrupt the interchain aggregation. Also, the solubility of these polymers was influenced by the molecular weights. So, in this work, we introduce alkyl chains to the backbone as side chains and control the molecular weights in appropriate range to ensure the final fully conjugated ladder polymers P3 and P6 own favorable solubility. Compound S1 is soluble in organic solvents. The Table 1 (in blow) show the solubility data of the polymers in our work.

Table 1. The solubility data of polymers in organic solvent

P1

P2

P3

P4

P5

P6

THF (mg/mL)

56

23

33

35

24

29

CHCl3 (mg/mL)

55

35

37

29

19

23

DMF (mg/mL)

24

17

20

20

13

16

Round 2

Reviewer 1 Report

I recommend the revised manuscript

Author Response

We thank the reviewer for the positive recommendation.

Reviewer 3 Report

Based on the clarification provided by the authors in the response letter, I believe the manuscript is worthy for publication.

Author Response

(The authors gave the same response as above.)
